# Gut-Microbiota Dysbiosis in Stroke-Prone Spontaneously Hypertensive Rats with Diet-Induced Steatohepatitis

**DOI:** 10.3390/ijms24054603

**Published:** 2023-02-27

**Authors:** Shini Kanezawa, Mitsuhiko Moriyama, Tatsuo Kanda, Akiko Fukushima, Ryota Masuzaki, Reina Sasaki-Tanaka, Akiko Tsunemi, Takahiro Ueno, Noboru Fukuda, Hirofumi Kogure

**Affiliations:** 1Division of Gastroenterology and Hepatology, Department of Medicine, Nihon University School of Medicine, 30-1 Oyaguchi-kamicho, Itabashi-ku, Tokyo 173-8610, Japan; 2Division of Nephrology, Hypertension and Endocrinology, Department of Medicine, Nihon University School of Medicine, 30-1 Oyaguchi-kamicho, Itabashi-ku, Tokyo 173-8610, Japan

**Keywords:** high-fat- and high-cholesterol-containing diet, *Firmicutes/Bacteroidetes* ratio, gut-microbiota, metabolic-dysfunction-associated fatty-liver disease, nonalcoholic steatohepatitis, small-intestinal bacterial overgrowth, stroke, hypertension

## Abstract

Metabolic-dysfunction-associated fatty-liver disease (MAFLD) is the principal worldwide cause of liver disease. Individuals with nonalcoholic steatohepatitis (NASH) have a higher prevalence of small-intestinal bacterial overgrowth (SIBO). We examined gut-microbiota isolated from 12-week-old stroke-prone spontaneously hypertensive-5 rats (SHRSP5) fed on a normal diet (ND) or a high-fat- and high-cholesterol-containing diet (HFCD) and clarified the differences between their gut-microbiota. We observed that the *Firmicute/Bacteroidetes (F/B)* ratio in both the small intestines and the feces of the SHRSP5 rats fed HFCD increased compared to that of the SHRSP5 rats fed ND. Notably, the quantities of the 16S rRNA genes in small intestines of the SHRSP5 rats fed HFCD were significantly lower than those of the SHRSP5 rats fed ND. As in SIBO syndrome, the SHRSP5 rats fed HFCD presented with diarrhea and body-weight loss with abnormal types of bacteria in the small intestine, although the number of bacteria in the small intestine did not increase. The microbiota of the feces in the SHRSP5 rats fed HFCD was different from those in the SHRP5 rats fed ND. In conclusion, there is an association between MAFLD and gut-microbiota alteration. Gut-microbiota alteration may be a therapeutic target for MAFLD.

## 1. Introduction

The number of patients with nonalcoholic fatty liver disease (NAFLD), including nonalcoholic steatohepatitis (NASH), has increased over the years [1]. As NASH causes cirrhosis and hepatocellular carcinoma (HCC), NASH is one of the important health issues worldwide. However, an unknown mechanism is also present in the pathogenesis of the development of NASH [2,3].

Fatty liver associated with metabolic dysfunction is common [4,5,6]. Metabolic-dysfunction-associated fatty-liver disease, “MAFLD,” may be a more appropriate overarching term [4,5]. Metabolic-dysfunction-associated fatty-liver disease is the principal worldwide cause of liver disease and affects nearly a quarter of the global population [4,5]. Diagnosis of MAFLD is based on the detection of liver steatosis together with the presence of at least one of three criteria that includes overweight or obesity, type 2 diabetes mellitus, or clinical evidence of metabolic dysfunction, such as an increased waist circumference and an abnormal lipid or glycemic profile [5]. Patients with hepatic steatosis and lean/normal weight is diagnosed as MAFLD in the presence of more than two metabolic risk abnormalities of the following criteria: an increased waist circumference, hypertension, an abnormal lipid or glycemic profile [5]. Patients with NAFLD are at a substantially higher risk of fatal and non-fatal cardiovascular events [6]. NAFLD and cardiovascular disease share multiple common conditions, such as obesity, diabetes, dyslipidemia and hypertension. These diseases may also share multiple common mechanisms, such as dietary habits, smoking, lack of exercise, gut-microbial dysbiosis, and genetics [6].

It has been reported that there is an association between NAFLD/NASH and gut-microbiota [5]. Individuals with NASH have a higher prevalence of small-intestinal bacterial overgrowth (SIBO) [7]. Intestinal mucosa-barrier malfunction may also play a role in NASH [8]. Individuals with NASH have a lower percentage of *Bacteroidetes* (*Bacteroidetes* total bacteria counts) than those with simple steatosis or healthy controls [9]. Thus, intestinal bacteria and gut-microbiota dysbiosis may play an important role in the development of NAFLD and NASH [3]. It has also been reported that gut-microbiota dysbiosis is linked to hypertension [10]. The gut-microbiota influence stroke pathogenesis and treatment outcomes [11,12].

Spontaneously hypertensive rats (SHR) and stroke-prone spontaneously hypertensive rats (SHRSP) are well-established parallel lines from outbred Wistar–Kyoto (WKY) rats [13,14]. We previously demonstrated a NASH model using arteriolipidosis-prone rats (ALR; SHRSP5), which are sublines obtained by the feeding of high-fat- and high-cholesterol-containing diets (HFCD) to SHRSP rats [15]. SHRSP5 rats fed HFCD possessed NASH, abnormal lipid, lean body, hypertension, and stroke [13,14,15].

In the present study, we examined the gut-microbiota isolated from stroke-prone spontaneously hypertensive-5 rats (SHRSP5) that were fed a normal diet (ND) or HFCD at 12 weeks of age and clarified the difference between their gut-microbiota. We observed differences between the microbiota of the feces in the SHRSP5 rats fed HFCD and those in the SHRP5 rats fed ND, as well as an increase in the *Firmicutes*/*Bacteroidetes* (*F/B*) ratio, which is a signature of gut dysbiosis, in the microbiota from the small intestine in the SHRSP5 rats fed HFCD. Our observation partially supports the concept of “MAFLD” from the point of view of gut-microbiota dysbiosis.

## 2. Results

### 2.1. Quantitative Analysis of 16S Ribosomal RNA Genes of Bacteria of Microbiota

Fecal-pellet DNA was isolated from the 12-week-old SHRSP5 rats fed ND or HFCD for 7 weeks [15]. As previously reported [15], in the HFCD group, pathological findings consistent with NASH were observed; however, in the ND group, only diffuse lipid droplets were seen in the hepatocytes at 12 weeks of age. As one rat died in the HFCD group, its fecal DNA could not be analyzed. At the same time, the DNA contents in the small intestines were also isolated from both groups of rats.

First, we performed real-time PCR to measure the 16S ribosomal (r)RNA genes of the bacteria in the small intestines and feces in both groups of rats (Table 1). We noticed that the quantities of the 16S rRNA genes in the small intestines of the SHRSP5 rats fed HFCD were significantly lower than those of the SHRSP5 rats fed ND (*p* < 0.05). However, the DNA from all samples were sufficient for the subsequent analysis.

Thus, HFCD reduced the 16S rRNA genes in the small intestines of the SHRSP5 rat, compared with ND. Interestingly, there may be an association between the reduction in bacteria and the fibrosis of the steatosis of the liver in the SHRSP5 rat fed HFCD. The effects of HFCD intake may be more important for the development of hepatic fibrosis in NASH than SIBO.

### 2.2. Next-Generation Sequencing of the V4–V5 Region of 16S rRNA Genes of Gut-Microbiota

Gut-microbiota dysbiosis is occasionally observed in patients with NASH [16]. The bacterial 16S rRNA gene has been used to define bacterial taxonomy and phylogeny. In order to understand the association between the gut-microbiota and the pathogenesis of NASH, we analyzed the V4–V5 region of the 16S rRNA from the bacteria in the small intestines and feces in the SHRSP5 rats fed ND or HFCD on the Illumina-MiSeq platform. The sequencing-read numbers are shown in Table 2. The sequence-read number ranged from 18,255 to 31,756. In the small intestines, the average sequence-read number of rats fed ND was similar to those of rats fed HFCD (28,819 ± 1944 vs. 29,567 ± 1956; no statistically significant difference). The coverage numbers were in a sequence around ~410 bp. These results indicate successful next-generation sequencing in the present study.

### 2.3. Microbiota of Small Intestine in SHRSP5 Rats Fed ND Are More Similar to Those of Small Intestine or Feces in SHRSP5 Rats Fed HFCD Than to Those of Feces in SHRSP5 Rats Fed ND

Next, we performed weighted UniFrac analyses to calculate the distances between the microbiota populations from the small intestines and feces in the SHRSP5 rats fed with a ND or HFCD [17] (Figure 1A). The microbiota of the small intestines in the SHRSP5 rats fed on ND were more similar to those of the small intestines or feces in the SHRSP5 rats fed on a HFCD than to those of the feces in the SHRSP5 rats fed with ND. The clustering analysis in the ß-diversity analysis of the microbiota populations also supported these results (Figure 1B,C).

A clear separation was observed in the principal-components analysis, clustering analysis, and ß-diversity analysis of the microbiota of the feces between the SHRSP5 rats fed on a HFCD and those fed on a ND (Figure 1A–C). Notably, the microbiota of the feces of the SHRSP5 rats fed an HFCD was different from those of the SHRSP5 rats fed an ND.

### 2.4. The Firmicutes/Bacteroidetes (F/B) Ratio Increased in the Small Intestines of SHRSP5 Rats Fed HFCD Compared to That in SHRSP5 Rats Fed ND

An increase in the *Firmicutes/Bacteroidetes* (*F/B*) ratio, caused by an expansion of *Firmicutes* and/or a contraction of *Bacteroidetes*, is considered a signature of gut dysbiosis [10]. The *F/B* ratio in the small intestines of the SHRSP5 rats fed an HFCD increased compared to that of the SHRSP5 rats fed an ND (Figure 2A). The *F/B* ratio in the feces of the SHRSP5 rats fed with the HFCD tended to increase compared to that of the SHRSP5 rats fed with the ND (Figure 2B).

The *F/B* ratio in the small intestines of the SHRSP5 rats fed with the HFCD was ~4.6-fold higher than that of the SHRSP5 rats fed the ND (Figure 2A). The *F/B* ratio in the feces of the SHRSP5 rats fed the HFCD tended to be ~1.7-fold higher than that of the SHRSP5 rats fed the ND (Figure 2B).

In both the small intestines and the feces of SHRSP5 rats fed on an HFCD, the number of both *Firmicutes* and *Bacteroidetes* decreased. In the feces of the SHRSP5 rats fed the HFCD, the number of *Proteobacteria* increased (Figure 3).

In the present study, among the *Firmicutes*, the *Allobaculum* decreased in the feces of the SHRSP5 rats fed with a HFCD. The *Lactobacillus* decreased and the *Streptococcus* increased in the small intestines of the SHRSP5 rats fed the HFCD. The *Clostridium* increased in both the small intestines and the feces of the SHRSP5 rats fed the HFCD. Of the *Bacteroides*, the *Porphyromonadaceae* decreased in feces of SHRSP5 rats fed the HFCD. Of the *Proteobacteria*, the *Escerichia* increased in both the small intestines and the feces of the SHRSP5 rats fed the HFCD.

## 3. Discussion

In the present study, we examined the gut-microbiota isolated from 12-week-old SHRSP5 rats fed a ND or a HFCD and clarified the differences between their gut-microbiota. We observed that the *F/B* ratio in both the small intestines and the feces of SHRSP5 rats fed the HFCD increased compared to that of the SHRSP5 rats fed the ND. Notably, the quantity of 16S rRNA genes in the small intestines of the SHRSP5 rats fed the HFCD were significantly lower than those of the SHRSP5 rats fed the ND. The microbiota of the feces of the SHRSP5 rats fed the HFCD was different from those of the SHRSP5 rats fed the ND.

Li et al. reported the ability of Grifola frondosa heteropolysaccharide to ameliorate NAFLD in rats fed a high-fat diet (HFD) and significantly increase the proportions of *Allobaculum* [18]. Increases in *Allobaculum* can help infant mice resist the development of obesity, according to an investigation of the intestinal microbiota in mice [19]. These reports partially support our observation that *Firmicutes* and *Allobaculum* decreased in the feces of the SHRSP5 rats fed the HFCD.

Panasevich et al. reported that soy protein is effective at preventing hepatic steatosis, and an analysis of fecal bacterial 16S rRNA revealed that soy-protein isolate intake elicited increases in *Lactobacillus* in obese Otsuka Long–Evans Tokushima fatty (OLETF) rats [20]. The rates of *Streptococcus* belonging to *Bacilli* were significantly increased in rats fed with a high-fat diet [21]. Compared with healthy subjects, NAFLD patients show an increase in the percentage of bacteria of pathogenic *Streptococcus* [22]. Previous studies [20,21,22] support our observations that the rates of *Lactobacillus* decreased and those of *Streptococcus* increased in the small intestines of the SHRSP5 rats fed the HFCD.

Individuals with NAFLD might be at increased risk of the development of *Clostridioides difficile* colitis [23]. *Clostridioides difficile* colitis can trigger changes associated with the development of NAFLD [24]. In our study, the *Clostridium* also increased in both small intestines and the feces of the SHRSP5 rats fed the HFCD.

High-fat diets result in quantitative alterations in the *aerobes* (*Escherichia coli*) in NASH rats [25]. Of the *Proteobacteria*, the *Escherichia* increased in both the small intestines and the feces of the SHRSP5 rats fed the HFCD. In 37.5% (12/32) of the patients with NAFLD, SIBO was present, with *Escherichia coli* as the predominant bacterium [26]. A previous study also demonstrated an increase in the *Escherichia genus* among gut-microbiota in the development and progression of NASH [27,28].

The presence of SIBO decreases small-intestinal movement in NASH rats [25]. A high-fat diet did not increase the *anaerobics (Lactobacilli)* [25]. *Bacteroides* species are also *anaerobic*. In the present study, of the *Bacteroides*, *Porphyromonadaceae* decreased in the feces of the SHRSP5 rats fed the HFCD. The presence of SIBO and endotoxemia can result in changes in toll-like receptor (TLR)-signaling gene expression, leading to the development of NAFLD [26]. The abundance of *Bacteroidetes phylum* may be increased, decreased, or unaltered in NASH patients [28].

Thus, SIBO plays a role in the development of NASH pathogenesis [7]. Patients with NASH and those with significant liver fibrosis on liver biopsy had a significantly higher incidence of SIBO than patients without NASH and those without significant liver fibrosis, respectively [29,30]. The onset of NASH in childhood is also a significant health problem [31]. There is an association between NAFLD and SIBO in obese children [32]. SIBO has an effect on the structural and functional characteristics of the liver, resulting in higher insulin and glucose levels, higher neutrophil-to-lymphocyte ratios, and a greater prevalence of NAFLD. A meta-analysis showed a possible association between SIBO and NAFLD in children [33].

The higher the grade of liver steatosis, the higher were the circulating lipopolysaccharide (LPS)-binding protein levels and SIBO rates seen in patients with morbid obesity and NAFLD [34]. The presence of SIBO may enhance intestinal permeability and endotoxemia in NASH patients [35]. Increased endotoxemia may enhance the innate immune response, including TLR-signaling pathways, as well as leading to inflammation and fat deposition in the liver.

The symptoms related to SIBO are bloating, diarrhea, malabsorption, body-weight loss, and malnutrition [36]. SIBO is a heterogeneous syndrome characterized by an increased number and/or abnormal type of bacteria in the small intestine [36]. Notably, the SHRSP5 rats fed with the HFCD presented diarrhea and body-weight loss compared to those fed with the ND [15]; these symptoms were consistent with those of SIBO. In the SHRSP5 rats fed the HFCD, abnormal types of bacteria were observed in the small intestines, although the number of these bacteria did not increase (Table 1). We noticed that the HFCD is more important for the development of hepatic fibrosis in NASH than SIBO.

High-fat diet (HFD)-dependent differences at the phylum, class, and genus levels appear to lead to dysbiosis, characterized by an increase in the *F/B* ratio, and *Firmicutes* was the dominant class in a male Sprague-Dawley (SD) rat (7 weeks old) fed HFD with steatohepatitis [37], supporting our observation (Figure 2B). An eight-week treatment of Gegen Qinlian decoction (GGQLD), a well-known traditional Chinese herbal medicine, improved these HFD-induced change [37]. Hugan Qingzhi tablet (HQT), which is a lipid-lowering and anti-inflammatory medicinal formula, has been used to prevent and treat NAFLD and reduced the abundance of the *F/B* ratio in HFD-fed rats [21]. Curcumin and metformin, which have a therapeutic effect against NAFLD, reduced the *F/B* ratio and reverted the composition of the HFD-disrupted gut-microbiota in male Sprague–Dawley rats fed HFD [38]. Gut-microbiota can play a role in the pathogenesis of NAFLD, as dysbiosis is associated with reduced bacterial diversity, altered F/B ratio, a relative abundance of alcohol-producing bacteria, or other specific genera [39].

Major risk factors of MAFLD are overweight/obesity, central obesity, type 2 diabetes mellitus, dyslipidemia, arterial hypertension, metabolic syndrome, insulin resistance, dietary factors, lifestyle, and sarcopenia [5]. It is known that gut-microbiota, hyperuricemia, hypothyroidism, sleep apnea syndrome, polycystic ovary syndrome, polycythemia, hypopituitarism, genetic and epigenetic factors, and family history of metabolic syndrome including high blood pressure are common and uncommon risk factors of MAFLD [5].

An association between hypertension and gut-microbiota alteration has been reported [10], as has an association between stroke and gut-microbiota alteration [11,12]. An association between obese and gut-microbiota alteration has also been reported [40], although fecal microbiota transplantation did not reduce body mass index. Evidence for the role of gut-microbiota in metabolic diseases including type 2 diabetes was provided [41]. Human and animal studies indicate the association between diets and hepatic steatosis [42,43]. The association between MAFLD and gut-microbiota alteration should now be clearer given the results of the present study. Dietary factors, such as high-calorie diets with rich saturated fats and cholesterol, soft drinks high in fructose, and highly processed foods, are known to influence the severity of NAFLD. Changing gut-microbiota also does so, at least in part [44]. In the present study, HFCD had an impact on changing gut-microbiota.

We observed an association between NASH and gut-microbiota alteration in the SHRSP5 rats, which originated from the stroke-prone, spontaneously hypertensive rats (SHRSP) fed the HFCD. The recent concept of MAFLD highlights the association between fatty liver disease, hypertension, stroke, and other metabolic diseases. The results from the present study may partially support the association between MAFLD and gut-microbiota alteration. Gut-microbiota alteration may be a therapeutic target for MAFLD. The real interest is how and why the altered microbiota are related to the pathological phenotype. Studies of the associated mechanism should be performed.

The 16S rRNA gene is present in multiple copies in the genomes of bacterial pathogens [45,46]. Therefore, amplicon-sequencing of the bacterium-specific 16S rRNA gene is a useful method for investigating a broad range of bacterial species. However, it is unclear whether the amplicon-sequencing-based detection of the 16S rRNA gene is useful for determining the causative pathogen. A major problem is that the 16S rRNA gene can be amplified not only for meaningful bacteria but also for meaningless bacteria, which is one of the limitations of this study.

Another limitation of the present study is that the number of rats used was small. This was because the present study was an initial study; we will elucidate the mechanisms further in a future study. For example, further improvement of bioinformatics and their analysis, the use of the QIME2 software, which uses amplicon sequence variant (ASV) instead of operational taxonomic unit (out) [47,48,49,50,51], or a denoising step, which allows for obtaining microbial taxa with a higher confidence [52], will be needed.

## 4. Materials and Methods

### 4.1. Animals

This investigation conformed to the Guide for the Care and Use of Laboratory Animals published by the US National Institutes of Health (NIH publication no. 85-23, 1996). The Ethics Committee of Nihon University School of Medicine examined all research protocols involving the use of animals and approved this study (no. 11-034). SHRSP5 rats were obtained from Disease Model Cooperative Research Association (Kyoto, Japan) [13,14]. The SHRSP5 rat is a subline obtained by feeding HFCD to SHRSP rats [53,54]. These SHRSP5 rats are characterized by fat deposition in their arteries, as well as fat deposition in and fibrosis of their livers, indicating the development of diet-induced NASH [15].

### 4.2. Dietary Intervention

The ND group was fed only a stroke-prone (SP) diet. SP diet was purchased as MF from Oriental Yeast Co., Ltd., Itabashi-ku, Tokyo, Japan. The HFCD consisted of 68% (*w*/*w*) SP diet, 25% (*w*/*w*) palm oil, 5% (*w*/*w*) cholesterol, and 2% (*w*/*w*) cholic acid [15]. In 100 g of ND, there was approximately 7.9 g water, 23.1 g protein, 5.1 g fat, 5.8 g ash, 2.8 g fiber, 55.3 g soluble without asphyxiation, and 359 kcal, according to the information from Oriental Yeast (https://www.oyc.co.jp/bio/LAD-equipment/LAD/ingredient.html (accessed on 13 February 2023)). The quantities of vitamins A, D3, E, K3, B1, B2, C, B6, B12, inositol, biotin, pantothenic acid, niacin, colin, and folic acid were 1283 IU, 137 IU, 9.1 mg, 0.04 mg, 2.05 mg, 1.1 mg, 4 mg, 0.87 mg, 5.5 mg, 439 mg, 27 μg, 2.45 mg, 10.61 mg, 0.18 g, and 0.17 mg, respectively, in 100 g of ND. We expected each rat to eat ~20 g of the diet daily. Experiments were conducted at least twice for consistent observations.

### 4.3. Sample Collection

Three rats from each group were examined [15]. Their feces were collected for 16S rRNA sequencing analysis. We only gathered the top layers of the feces and performed the isolation under sterile conditions to avoid bacterial contamination. Isoflurane was used as an anesthesia method for sampling the contents of small intestines. Heart blood was collected under general anesthesia; after abdominal median incision, heart blood was collected as described elsewhere [15]. After the incision of perianal, we collected the content of small intestine for further analysis. We performed animal experiments according to the Japanese animal welfare guidelines (https://www.maff.go.jp/j/chikusan/sinko/animal_welfare.html (accessed on 13 February 2023)) at that time.

### 4.4. Quantification of 16S rRNA Genes by Real-Time PCR

The total bacterial genomic DNA was extracted using the Extrap Soil DNA Kit Plus ver.2 (Nippon Steel Corporation, Tokyo, Japan) and stored at −20 °C prior to further analysis. The DNA was used in equal amounts for further PCR analysis.

The total number of bacterial 16S rRNA genes was estimated using a TaqMan-based qPCR approach with primers Bac1055YF, Bac1392R, and Q-probe Bac 1115Probe, which were described previously [55] (Table 3).

### 4.5. Next-Generation Sequencing 16S rRNA Genes

In general, 16S and/or internal transcribed spacer ribosomal RNA sequencing are performed for the amplicon sequencing methods to identify and compare the flora of bacteria or fungus of collected samples [50]. This method could identify them after concentrating the original materials using the next-generation sequencing. We performed sequencing analysis of 16S rRNA genes in the present study.

The PCR with high-fidelity-DNA polymerase was used to amplify the V4–V5 region of the 16S rRNA gene with primers U515F and 926R (Table 3). Agilent 2100 bioanalyzer (Agilent technologies, Santa Clara, CA, USA) and PicoGreen dsDNA Assay Kit (Invitrogen, Carlsbad, CA, USA) were used to purify and quantify the resulting PCR amplicons. The Illumina-MiSeq platform (Illumina, San Diego, CA, USA) was used to pool the amplicons in equal amounts and implement the paired-end 2 × 250-base-pair sequencing. Finally, base-pair sequences of ~410 bp were analyzed.

### 4.6. Data Analysis

Standard bioinformatics-alignment comparison was utilized for data analysis [56]. The Quantitative Insights Into Microbial Ecology (QIIME) pipeline was employed to process the sequencing data [16]. Paired-end reads were demultiplexed according to a combination of forward and reverse indices. Additional quality filtering included exact match to sequencing primers and an average quality score of 30 or higher on each read. Prior to further analysis, each paired-end read was stitched into one contiguous read using the fast length adjustment of short reads (FLASH) software tool. Reads that could not be joined were excluded from downstream analysis. All sequences passing filters were aligned against a Silva non-redundant 16S reference database (v108) and assigned taxonomic classifications using USEARCH at a 97% identity threshold. Dereplication to unique reference-sequence-based operational taxonomic units (refOTU) was performed using UCLUST at a 97% clustering threshold and summarized in a refOTU table. Additional alpha-diversity measures and normalized-per-level taxonomic abundances were created using custom scripts written in R [10]. Differentially significant features at each level were identified using linear discriminant analysis (LDA), along with effect-size measurements (LEfSe) [57]. Three-dimensional principal-coordinates analysis (PCoA) plots using the tree-based UniFrac distance metric were generated through custom scripts in R and scripts from the QIIME package [16]. The OUT taxonomic classification was conducted by BLAST, searching the representative sequences set against the database using the best hit, as in previous studies [58]. Classification of bacterial taxonomy based on the end product was performed as previously described [59]. Briefly, genera were classified into more than one group if they were defined as producers of multiple metabolites. Genera that were defined as producing equol, histamine, hydrogen, and propionate constituted only a minor portion of the population and were therefore excluded from this analysis. A representative sequence from each OTU was selected according to the default parameters.

## 5. Conclusions

As in SIBO syndrome, the SHRSP5 rats fed with a HFCD presented diarrhea and body-weight loss with abnormal types of bacteria in their small intestines, although the number of these bacteria did not increase. Our results strongly support the association between MAFLD and gut-microbiota alteration.

## Figures and Tables

**Figure 1 ijms-24-04603-f001:**
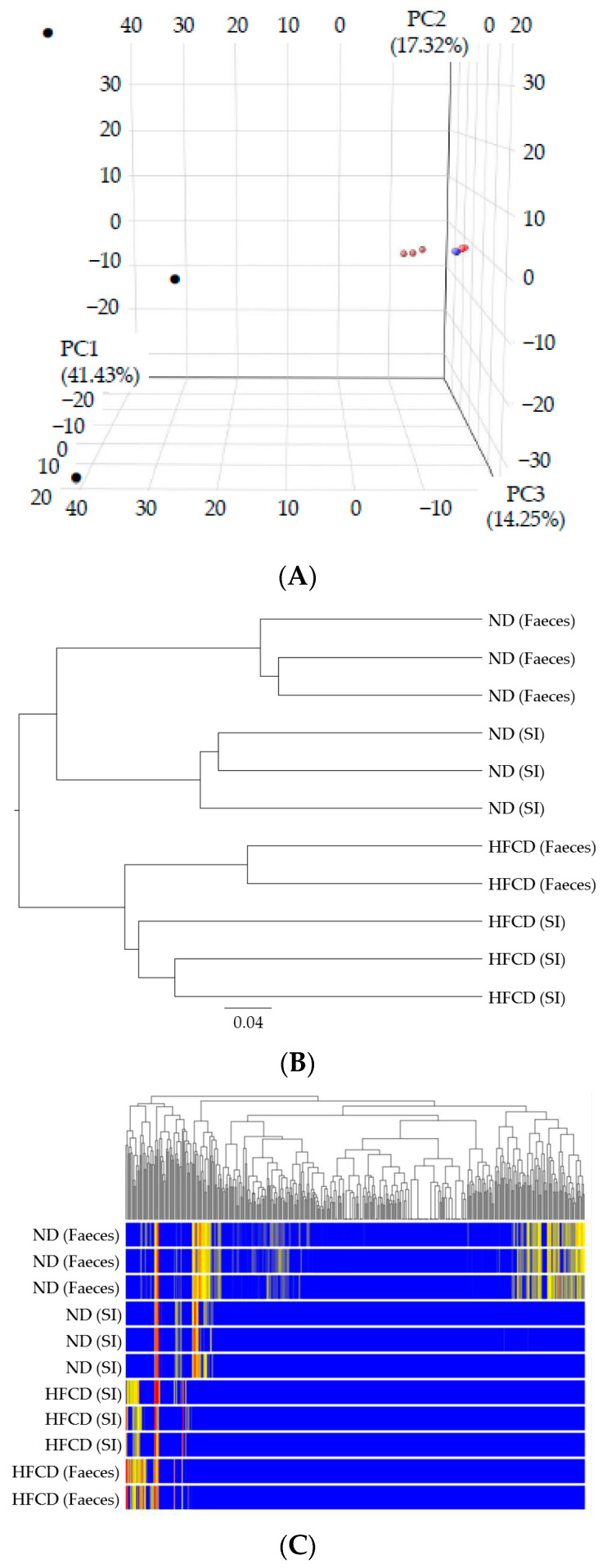
Principal-components analysis and clustering analysis in ß-diversity analysis. (**A**) Principal-components analysis of microbiota populations of small intestines and feces in stroke-prone spontaneously hypertensive-5 (SHRSP5) rats fed normal diet (ND) or high-fat- and high-cholesterol-containing diet (HFCD) by plotted unweighted UniFrac distances [15,17]. Microbiota of small intestine in SHRSP5 rats fed ND were more similar to those of small intestines or feces in SHRSP5 rats fed HFCD than to those of feces in SHRSP5 rats fed ND. Small intestine (SI) in SHRSP5 rats fed ND (small brown circles, *n* = 3), feces in SHRSP5 rats fed ND (small black circles, *n* = 3), small intestines in SHRSP5 rats fed HFCD (small red circles, *n* = 3), and feces in SHRSP5 rats fed HFCD (small blue circles, *n* = 2). (**B**) Clustering analysis in ß-diversity analysis of microbiota populations of small intestines and feces in SHRSP5 rats fed ND or HFCD. (**C**) Results of clustering analysis. Data were analyzed using GeneSpring GX software. Microbiota of feces in SHRSP5 rats fed ND belonged to different groups.

**Figure 2 ijms-24-04603-f002:**
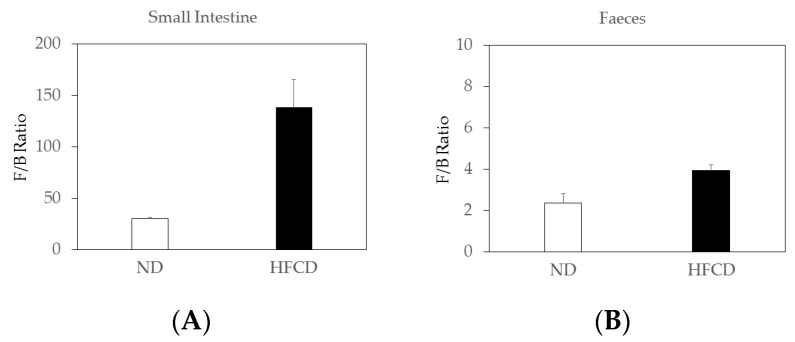
*Firmicute*-to-*Bacteroidetes* ratio (*F/B* ratio) was calculated from the sequence-read number as a biomarker of gut dysbiosis. Comparison of microbiota composition in small intestines (**A**) and feces (**B**) in stroke-prone spontaneously hypertensive-5 (SHRSP5) rats fed normal diet (ND) or high-fat- and high-cholesterol-containing diet (HFCD).

**Figure 3 ijms-24-04603-f003:**
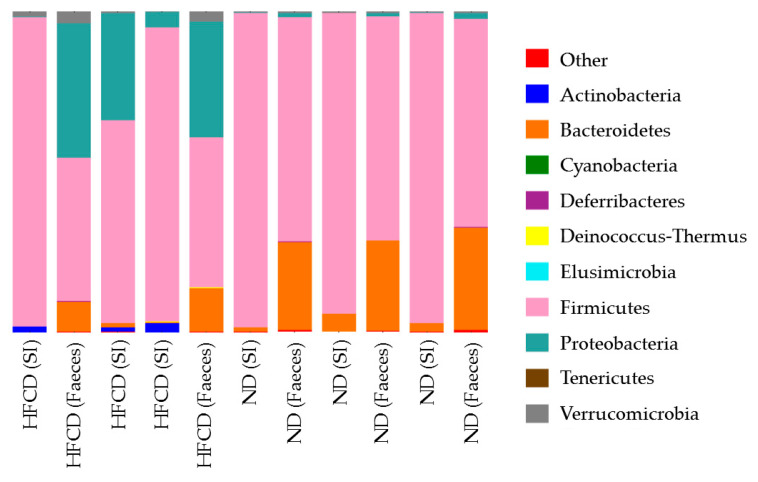
Description of proportional values of indicated phyla of microbiota from each sample. Comparison of microbiota composition in small intestines and feces in stroke-prone spontaneously hypertensive-5 (SHRSP5) rats fed normal diet (ND) or high-fat- and high-cholesterol-containing diet (HFCD). SI, small intestine.

**Table 1 ijms-24-04603-t001:** Quantitative analysis of 16S rRNA genes of bacteria of small intestine (SI) and feces in stroke-prone spontaneously hypertensive-5 (SHRSP5) rats fed normal diet (ND) or high-fat- and high-cholesterol-containing diet (HFCD) by real-time PCR.

Materials	16S rRNA Genes (Copies/g)
ND (SI)	4.8 × 10^10^
ND (SI)	5.1 × 10^10^
ND (SI)	8.0 × 10^10^
ND (Feces)	5.0 × 10^11^
ND (Feces)	5.2 × 10^11^
ND (Feces)	6.4 × 10^11^
HFCD (SI)	8.9 × 10^6^
HFCD (SI)	4.3 × 10^7^
HFCD (SI)	2.0 × 10^8^
HFCD (Feces)	2.3 × 10^10^
HFCD (Feces)	5.2 × 10^10^

**Table 2 ijms-24-04603-t002:** Sequence-read number of the V4–V5 region of the 16S rRNA from bacteria of small intestines (SIs) and feces in stroke-prone spontaneously hypertensive-5 (SHRSP5) rats fed normal diet (ND) or high-fat- and high-cholesterol-containing diet (HFCD) on the lllumina MiSeq platform.

Materials	Sequence-Read Number
ND (SI)	31,756
ND (SI)	27,989
ND (SI)	28,957
ND (Feces)	18,255
ND (Feces)	24,888
ND (Feces)	21,820
HFCD (SI)	29,786
HFCD (SI)	26,581
HFCD (SI)	30,091
HFCD (Feces)	29,040
HFCD (Feces)	27,399

**Table 3 ijms-24-04603-t003:** The PCR primers and probe used to quantify and amplify total bacterial 16S rRNA genes.

Primer/Probe	Sequence	Target Gene
Bac1055YF *	5′-ATGGYTGTCGTCAGCT-3	Bacteria
Bac1392R *	5′-ACGGGCGGTGTGTAC-3	Bacteria
Bac1115Probe *	5′-FAM-CAACGAGCGCAACCC-TAMRA	Bacteria
U515F	5′-GTGYCAGCMGCCGCGGTA-3′	V4–V5 region of the 16S rRNA
926R	5′-CCGYCAATTCMTTTRAGTT-3′	V4–V5 region of the 16S rRNA

* Reference [53].

## Data Availability

The data underlying this article are available in this article.

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
