# Peer review of "Gut-Microbiota Dysbiosis in Stroke-Prone Spontaneously Hypertensive Rats with Diet-Induced Steatohepatitis"

_ijms, 2023, doi:10.3390/ijms24054603_

Round 1

Reviewer 1 Report

The novelty and the quality of the manuscript are good and it does not need extensive improvement before publication. It is carefully organized and written. It is easy to follow it and contains clear comments and conclusions.  In my opinion, this manuscript is very detailed and meticulous, it covers all the literature in the field with critical point of view. The topic have been completely covered and is well connected through the text. There is a significant  novelty in presented topic.  For all these reasons, I can only recommend the acception of the manuscript after minor revision:

 1. I think that part 2.1. Quantitative analysis of 16S rRNA genes of bacteria of microbiota in the present study could be extended, more examples  should be added. This would be valuable for later publication citation.

 2. The superiority of  the association between MAFLD and gut-microbiota alteration than other correlations should be more emphasized.

 3. The manuscript should be extended in scientific discussion. The authors presented their results and compared to some works, but did not present explanations for the reasons to reach these results.

 4. Not all of the described results are covered in the discussion section

 5. No all information was given on the animal's diet - whether it was a semi-synthetic or a farm diet, what was its composition, what were the caloric, fiber and vitamin C contents,

Author Response

Response to the Reviewer 1:

Thank you very much for your valuable comments.

Response to your comment 1: “I think that part 2.1. Quantitative analysis of 16S rRNA genes of bacteria of microbiota in the present study could be extended, more examples should be added. This would be valuable for later publication citation.”

Thank you very much for your valuable comments. We agree with you. We revised the result section accordingly.

Response to your comment 2: “The superiority of the association between MAFLD and gut-microbiota alteration than other correlations should be more emphasized.”

Thank you very much for your valuable comments. We agree with you. Accordingly, we revised our manuscript as follows.

In Discussion section, pages 8-9,

…[39]. Major risk factors of MAFLD are overweight/obesity, central obesity, type 2 diabetes mellitus, dyslipidemia, arterial hypertension, metabolic syndrome, insulin resistance, dietary factors, lifestyle and sarcopenia [5]. It is known that gut-microbiota, hyperuricemia, hypothyroidism, sleep apnea syndrome, polycystic ovary syndrome, polycythemia, hy-popituitarism, genetic and epigenetic factors and family history of metabolic syndrome including high blood pressure are common and uncommon risk factors of MAFLD [5].

An association between hypertension and gut-microbiota alteration has been report-ed [10] as has an association between stroke and gut-microbiota alteration [11,12]. An association between obese and gut-microbiota alteration has also been reported[40] although fecal microbiota transplantation did not reduce body mass index. Evidence for the role of gut-microbiota in metabolic diseases including type 2 diabetes was provided [41]. Human and animal studies indicate the association between diets and hepatic steatosis [42,43]. The association between MAFLD and gut-microbiota alteration should be focused from the results of the present study. Dietary factors, such as high-calorie diets with rich saturated fats and cholesterol, soft drinks high in fructose and highly processed foods, are known to influence the severity of NAFLD. Changing gut-microbiota also does so at least in part [44]. In the present study, HFCD had an impact on the changing gut-microbiota.

We observed an association between NASH and gut-microbiota alteration in the SHRSP5 rats, which originated from the stroke-prone, spontaneously hypertensive rats fed the HFCD. The recent concept of MAFLD highlights the association between fatty liver disease, hypertension, stroke and other metabolic diseases. The results from the present study may partially support the association between MAFLD and gut-microbiota alteration. Gut-microbiota alteration may be a therapeutic target for MAFLD. The real interest is how and why the altered microbiota are related to the pathological phenotype. Studies of the associated mechanism should be performed.

The 16S rRNA gene is present in multiple copies in the genomes of bacterial pathogens [45,46]. Therefore, amplicon-sequencing of the bacterium-specific 16S rRNA gene is a useful method for investigating a broad range of bacterial species. . However, it is unclear whether the amplicon-sequencing-based detection of the 16S rRNA gene is useful for determining the causative pathogen. A major problem is that the 16S rRNA gene can be amplified not only for those of meaning bacteria but also for those of meaningless bacteria, which is one of the limitations of this study.

Another limitation of the present study is that the number of rats used is small because the present study is an initial study and we will elucidate the further mechanism in the future study. Further improvement of bioinformatics and their analysis, for example, the use of the QIME2 software which uses amplicon sequence variant (ASV) instead of operational taxonomic unit (OTU) [47–51] or denoising step which allow for obtaining microbial taxa with a higher confidence [52], should be needed.

Response to your comment 3: “The manuscript should be extended in scientific discussion. The authors presented their results and compared to some works, but did not present explanations for the reasons to reach these results.”

Thank you very much for your valuable comments. We agree with you. We extensively revised our manuscript accordingly.

Response to your comment 4: “Not all of the described results are covered in the discussion section.”

Thank you very much for your valuable comments. We agree with you. We extensively revised our manuscript accordingly.

Response to your comment 5: “No all information was given on the animal's diet - whether it was a semi-synthetic or a farm diet, what was its composition, what were the caloric, fiber and vitamin C contents,”

Thank you very much for your valuable comments. We agree with you. Accordingly, we extensively revised our manuscript as follows.

In Materials and Methods section, page 9,

4.2. Dietary Intervention

The ND group was fed only a stroke-prone (SP) diet. SP diet was purchased as MF from Oriental Yeast Co., Ltd, Itabashi-u, Tokyo, Japan. The HFCD consisted of 68% (w/w) SP diet, 25% (w/w) palm oil, 5% (w/w) cholesterol and 2% (w/w) cholic acid [15]. Total of 100 g ND included approximately 7.9g water, 23.1 g protein, 5.1 g fat, 5.8 g ash, 2.8 g fiber, 55.3 g soluble without asphyxiation and 359 kcal, according to the information from  Oriental Yeast (https://www.oyc.co.jp/bio/LAD-equipment/LAD/ingredient.html). The quantities of vitamins A, D3, E, K3, B1, B2, C, B6, B12, inositol, biotin, pantothenic acid, niacin, colin and folic acid were 1283 IU, 137 IU, 9.1 mg, 0.04 mg, 2.05 mg, 1.1 mg, 4 mg, 0.87 mg, 5.5 mg, 439 mg, 27 μg, 2.45 mg, 10.61 mg, 0.18 g and 0.17 mg, respectively, in the 100 g ND. We expected each rat ate ~20 g diet daily. Experiments were conducted at least twice for consistent observations.

Reviewer 2 Report

The present study mainly focused on gut microbiota alteration in SHRSP rats with high fat and cholesterol (HFCD) compared to the normal diet (stroke-prone diet, SP). Although the authors observed a sort of gut microbiota change after HFCD treatment in SHRSP rats, such as the increase of the Firmicutes/Bacteroides (F/B) ratio, reduction of 16S rRNA-based genes quantity, etc., we only acquired some basic information from this study, rather than any essential information regarding causality or mechanisms. We commonly know that food, especially high-fat-based food, can easily change gut microbiota more than other factors, like genetic factors. Therefore, undoubtedly HFCD administration can change the gut microbiota. But, the real interest is how and why the altered microbiota are related to the pathological phenotype. Overall, in the reviewer’s opinion, only limited information can be given through the current results from the present animal study. The related mechanism studies must be performed as possible. 

Author Response

Response to the Reviewer 2:

Thank you very much for your valuable comments.

Response to your comments: “But, the real interest is how and why the altered microbiota are related to the pathological phenotype. Overall, in the reviewer’s opinion, only limited information can be given through the current results from the present animal study. The related mechanism studies must be performed as possible.”

Thank you very much for your valuable comments. We agree with you. Accordingly, we extensively revised our manuscript and added the limitation of study as follows.

In Discussion section, pages 8-9,

… Gut-microbiota alteration may be a therapeutic target for MAFLD. The real interest is how and why the altered microbiota are related to the pathological phenotype. Studies of the associated mechanism should be performed.

The 16S rRNA gene is present in multiple copies in the genomes of bacterial pathogens [45,46]. Therefore, amplicon-sequencing of the bacterium-specific 16S rRNA gene is a useful method for investigating a broad range of bacterial species. . However, it is unclear whether the amplicon-sequencing-based detection of the 16S rRNA gene is useful for determining the causative pathogen. A major problem is that the 16S rRNA gene can be amplified not only for those of meaning bacteria but also for those of meaningless bacteria, which is one of the limitations of this study.

Another limitation of the present study is that the number of rats used is small because the present study is an initial study and we will elucidate the further mechanism in the future study. Further improvement of bioinformatics and their analysis, for example, the use of the QIME2 software which uses amplicon sequence variant (ASV) instead of operational taxonomic unit (OTU) [47–51] or denoising step which allow for obtaining microbial taxa with a higher confidence [52], should be needed.

Reviewer 3 Report

The manuscript is valuable, it concerns the important and often occurring disease, and it was prepared with the use of modern methods.

I have some suggestions:

In the abstract, please delete or rewrite the sentence:” We previously demonstrated a NASH model using arteriolipidosis-prone rats (ALR; SHRSP5), which are sublines obtained by feeding a high-fat- and high-cholesterol-containing diet (HFCD) to stroke-prone spontaneously hypertensive rats (SHRSP).” The previous findings should not appear in the abstract of a new study. Alternatively, you can rewrite the abstract explaining that the current study was made with the use of previously obtained experimental model.

It is not clear for a reader how microbiota composition was determined. Please explain it more clearly. Such explanation is missing in Material and methods section, it is also difficult to understand that reading Results or Discussion.

There is a mess in the style of writing of  taxonomic names in all manuscript. Latin names for species and  genus should be written in italics, please check the text carefully.

Author Response

Response to the Reviewer 3:

Thank you very much for your valuable comments.

Response to your comment 1: In the abstract, please delete or rewrite the sentence: ”We previously demonstrated a NASH model using arteriolipidosis-prone rats (ALR; SHRSP5), which are sublines obtained by feeding a high-fat- and high-cholesterol-containing diet (HFCD) to stroke-prone spontaneously hypertensive rats (SHRSP).” The previous findings should not appear in the abstract of a new study. Alternatively, you can rewrite the abstract explaining that the current study was made with the use of previously obtained experimental model.”

Thank you for your valuable suggestions. According to your suggestion, we deleted this sentence.

Response to your comment 2: “It is not clear for a reader how microbiota composition was determined. Please explain it more clearly. Such explanation is missing in Material and methods section, it is also difficult to understand that reading Results or Discussion.”

Thank you for your valuable suggestions. We agree with you. According to your suggestion, we extensively revised Material and methods, Results and Discussion section of our revised manuscript.

Response to your comment 3: “There is a mess in the style of writing of  taxonomic names in all manuscript. Latin names for species and  genus should be written in italics, please check the text carefully.”

Thank you for your valuable suggestions. We agree with you. According to your suggestion, we made corrections.

Reviewer 4 Report

1.  Bacterial names require italics.

2. In keywords: all abbreviations presents the full names.
3. In 4.1 Animal: SHRSP5 rats or SHRSP rats?

4. Please add the description of sample collections.

5. Please add the description of animal welfare.

Author Response

Response to the Reviewer 4:

Thank you very much for your valuable comments.

Response to your comment 1: “Bacterial names require italics.”

Thank you for your valuable suggestions. We agree with you. According to your suggestion, we made corrections.

Response to your comment 2: “In keywords: all abbreviations presents the full names.”

Thank you for your valuable suggestions. We agree with you. According to your suggestion, we made corrections.

Response to your comment 3: “In 4.1 Animal: SHRSP5 rats or SHRSP rats?”

Thank you for your valuable suggestions. “SHRSP5 rats” is right. According to your suggestion, we made corrections.

Response to your comment 4: “Please add the description of sample collections.”

Thank you for your valuable suggestions. We agree with you. According to your suggestion, we revised our manuscript as follows.

4.3. Sample Collection

Three rats from each group were examined [15]. Their feces were collected for 16S rRNA sequencing analysis. We only gathered the top layers of the feces and performed the isolation under sterile conditions to avoid bacterial contamination. Isoflurane was used as an anesthesia method for sampling the contents of small intestines. Heart blood was collected Under the general anesthesia, after abdominal median incision, heart blood was collected as described elsewhere [15]. After the incision of perianal,  we collected the content of small intestine for further analysis. We performed animal experiments according to the Japanese animal welfare guideline (https://www.maff.go.jp/j/chikusan/sinko/animal_welfare.html), at that time.

Response to your comment 5: “Please add the description of animal welfare.”

Thank you for your valuable suggestions. We agree with you. According to your suggestion, we revised our manuscript as follows.

4.3. Sample Collection

Three rats from each group were examined [15]. Their feces were collected for 16S rRNA sequencing analysis. We only gathered the top layers of the feces and performed the isolation under sterile conditions to avoid bacterial contamination. Isoflurane was used as an anesthesia method for sampling the contents of small intestines. Heart blood was collected Under the general anesthesia, after abdominal median incision, heart blood was collected as described elsewhere [15]. After the incision of perianal, we collected the content of small intestine for further analysis. We performed animal experiments according to the Japanese animal welfare guideline (https://www.maff.go.jp/j/chikusan/sinko/animal_welfare.html), at that time.

Reviewer 5 Report

The paper by Kanezawa et al. reports a metataxonomics analysis of 12-week-old stroke-prone spontaneously hypertensive-5 rats fed on normal and HFCD diets. 

I personally think that the paper requires a restyling.

The introduction is too much fragmented and the sentences require to be better connected.

Also the results need to be better presented. For instance, the different taxonomic levels (the statistically significant ones) deserve a separate figure or table.

The number of rats used is low. Please justify why the number is low.

I personally encourage the authors to run the analyses by using the QIIME2 software, which uses ASV instead of OTU. Also in this new release has a denoising step which allow for obtaining taxa with a higher confidence. Bioinformatic details are totally missing.

It's not cler why the authors decided to plot only taxa distribution at the phylum level, without reporting statistically significances that marked the compared groups.

Generally I think that all the obtained results need to be better reported and explained.

Author Response

Response to the Reviewer 5:

Thank you very much for your valuable comments.

Response to your comment 1: “I personally think that the paper requires a restyling.”

Thank you for your valuable suggestions. We agree with you. According to your suggestion, we extensively revised our manuscript.

Response to your comment 2: “The introduction is too much fragmented and the sentences require to be better connected.”

Thank you for your valuable suggestions. We agree with you. According to your suggestion, we revised our manuscript.

Response to your comment 3: “Also the results need to be better presented. For instance, the different taxonomic levels (the statistically significant ones) deserve a separate figure or table.”

Thank you for your valuable suggestions. As the number of rats we analyzed is small, we also described the limitation of study.

In Result section, page 3,

…The sequencing-read numbers are shown in Table 2. The number of sequence-read ranged from 18,233 to 31,756. In the small intestines, the average number of sequence-read of rats fed ND was similar to those of rats fed HFCD (28,819±1944 vs. 29,567±1956; not statistically significant difference). The coverage numbers were in a sequence around ~410 bp. These results also mean the successful next-generation sequencing in the present study.

In Discussion section, pages 8-9,

…… Gut-microbiota alteration may be a therapeutic target for MAFLD. The real interest is how and why the altered microbiota are related to the pathological phenotype. Studies of the associated mechanism should be performed.

The 16S rRNA gene is present in multiple copies in the genomes of bacterial pathogens [45,46]. Therefore, amplicon-sequencing of the bacterium-specific 16S rRNA gene is a useful method for investigating a broad range of bacterial species. . However, it is unclear whether the amplicon-sequencing-based detection of the 16S rRNA gene is useful for determining the causative pathogen. A major problem is that the 16S rRNA gene can be amplified not only for those of meaning bacteria but also for those of meaningless bacteria, which is one of the limitations of this study.

Another limitation of the present study is that the number of rats used is small because the present study is an initial study and we will elucidate the further mechanism in the future study. Further improvement of bioinformatics and their analysis, for example, the use of the QIME2 software which uses amplicon sequence variant (ASV) instead of operational taxonomic unit (OTU) [47–51] or denoising step which allow for obtaining microbial taxa with a higher confidence [52], should be needed.

Response to your comment 4: “The number of rats used is low. Please justify why the number is low.”

Thank you for your valuable suggestions. “Another limitation of the present study is that the number of rats used is small because the present study is an initial study and we will elucidate the further mechanism in the future study.”

Response to your comment 5: “I personally encourage the authors to run the analyses by using the QIIME2 software, which uses ASV instead of OTU. Also in this new release has a denoising step which allow for obtaining taxa with a higher confidence. Bioinformatic details are totally missing. It's not cler why the authors decided to plot only taxa distribution at the phylum level, without reporting statistically significances that marked the compared groups.”

Thank you for your valuable suggestions. We described the limitation of study, regarding this point. Another limitation of the present study is that the number of rats used is small because the present study is an initial study and we will elucidate the further mechanism in the future study. According to your suggestion, we also mentioned as follows.

In Discussion section, pages 8-9,

…… Gut-microbiota alteration may be a therapeutic target for MAFLD. The real interest is how and why the altered microbiota are related to the pathological phenotype. Studies of the associated mechanism should be performed.

The 16S rRNA gene is present in multiple copies in the genomes of bacterial pathogens [45,46]. Therefore, amplicon-sequencing of the bacterium-specific 16S rRNA gene is a useful method for investigating a broad range of bacterial species. . However, it is unclear whether the amplicon-sequencing-based detection of the 16S rRNA gene is useful for determining the causative pathogen. A major problem is that the 16S rRNA gene can be amplified not only for those of meaning bacteria but also for those of meaningless bacteria, which is one of the limitations of this study.

Another limitation of the present study is that the number of rats used is small because the present study is an initial study and we will elucidate the further mechanism in the future study. Further improvement of bioinformatics and their analysis, for example, the use of the QIME2 software which uses amplicon sequence variant (ASV) instead of operational taxonomic unit (OTU) [47–51] or denoising step which allow for obtaining microbial taxa with a higher confidence [52], should be needed.

Response to your comment 6: “Generally I think that all the obtained results need to be better reported and explained.”

Thank you for your valuable suggestions. We agree with you. We extensively revised our manuscript.

Round 2

Reviewer 2 Report

The authors try to clarify the high-fat diet (HFD) perturbing the gut microbiota in stroke-prone spontaneously hypertensive rats. But it is well-known that HFD can easily change the gut microbiota. In order to investigate the special difference, the wild-type rats fed a regular diet or HFD should be additionally compared. In addition, the data for testifying phenotype also must be provided, like related serum or protein parameters and ECG, etc. Only 16s rRNA metagenomics data is strictly limited without fully understanding the potential reason. Thus, we suggest that the authors need not only add the animal experiments, but also the corresponding phenotype data. If the authors want to improve the quality of the study, the study of the associated mechanisms is also essential.

Reviewer 5 Report

After the made corrections, the paper is ready to be published.